

# Yield-limiting nutrient response of lowbush blueberry grown in recent and ancient alluvial soils of the Mekong Delta

Ngo Phuong Ngoc, Le Minh Ly and Pham Thi Phuong Thao

Department of Plant Physiology-Biochemistry, College of Agriculture, Can-Tho University, Can Tho, Vietnam

## ABSTRACT

**Background.** Blueberries are ornamental plants grown in pots in many yards in the Mekong Delta (MD) region. In this region, the recent alluvial (RA) soil is fertile and ancient alluvial (AA) soil is considered degraded because it only has around a quarter of the nutrient content of the RA soil. Both soils have a high clay content, so organic matter is needed to improve their physical condition. This study aimed to identify the nutrients that limit the yield of blueberries in RA and AA soils of the MD.

**Methods.** The pot experiment was performed using a factorial randomized block design (RBD) with two factors: (a) two soil types (RA and AA) and (b) four omission or treatment conditions (NPK, PK, NK, and NP). The same fertilizer formula was used for all treatments, including $45N–20P_2O_5–20K_2O$ and mixing CHC (10 tha$^{-1}$) into the potting soil.

**Results.** The blueberry yield in AA soil was only 81% of that in RA soil. In both RA and AA soils, N omission caused foliar N content deficiency (10.42 g kg$^{-1}$), resulting in the content of foliar P (0.84 g kg$^{-1}$) and K (3.78 g kg$^{-1}$) to fall below the Trevett threshold. In both RA and AA, N omission resulted in reduced fruit yield (47% and 39%, respectively) as well as reduced weight of the stem (70% and 42%, respectively) and leaf (59% and 46%, respectively). Increased crop yields in soils were mainly related to nitrogen fertilizer. The indigenous nutrient supply (INS) of RA, which is fertile, was high but its apparent nutrient recovery efficiency (ARE) index was low, whereas the INS of AA, or the level of degraded soil, was low but its ARE index was high. In alluvial soils, the higher the INS level, the less positive the impact on the ARE index. In AA soil, the indigenous N and K supplies can be improved through fertilizer investment; however, a balance must be achieved considering economic efficiency.

## INTRODUCTION

Many yards in the Mekong Delta (MD) grow blueberries in pots (*Center of Agricultural Crops, 2023*). Blueberries have a lot of species and varieties. Some are shade-torelant understory species. Blueberries require soil with low clay content for good drainage and a pH between 4.5–5 (*Mississippi State University Extension Service, 2023*).

Corresponding author
Ngo Phuong Ngoc,
npngoc@ctu.edu.vn

There are two types of alluvial sediments in the MD: recent alluvium (RA) and ancient alluvium (AA). The fertility of AA soil is lower than that of RA soil; however, both soil types have a high clay content (*Van Dang et al., 2021*), resulting in poor drainage (*Tran, 2004*; *Quang, 2013*). Therefore, a certain amount of organic matter is required to improve soil under clay-rich conditions (*Brunswick, 2011*).

Fertilizer recommendations for lowbush blueberries are primarily based on leaf tissue testing because nutrient concentrations in the soil are not always the same as those in the leaves; therefore, soil testing is only supplementary (*Yarborough & Smagula, 2013*). This study used the published sufficiency ranges of N, P, and K in lowbush blueberry leaves (*Trevett, 1972*) to assess the nutritional status of the soil.

The efficiency of inorganic fertilizer recovery from plants is often low in many soil types because of leaching, ammonia volatilization, or immobilization (*Le Lafond, 2022*; *Vincent, 2022*). Especially in tropical areas with a soil pH below 5.5, P precipitated by Al, Fe, and P ions is rapidly fixed by the soil matrix (*Johan et al., 2010*; *Karthik & Maheswari, 2021*).

The omission plot method is commonly used to evaluate yield-limiting nutrients based on crop responses to fertilizers (*Shiveshwar et al., 2021*). In this approach, the total amount of N, P, or K uptake by the plant in the omission treatment is considered the supply capacity of $N$ (INS), P (IPS), and K (IKS) (*Congreves et al., 2021*). Data on the supply capacity of soil can guide fertilizer recommendations (*Dobermann et al., 2003*). These data can be obtained from pot experiments under controlled conditions in laboratory and greenhouse studies.

Several studies have been conducted on the response of blueberries to N, P, and K fertilizers, mainly in in situ weathered sandy soils. In this study, two high-clay soils with different soil fertility were hypothesized to potentially differ in soil-supplied mineral composition and crop productivity respond differently to the soil fertility. Therefore, this study aimed to identify yield-limiting nutrients in the soils of blueberry plants (*Vaccinium tenellum*) grown in AA and RA soils in the MD using a greenhouse pot experiment.

# MATERIALS & METHODS

## Study site parameters

### Study site and soil

Survey and soil sampling for the greenhouse experiment was carried out in April 2021. The two soils sampled for the greenhouse experiment were: RA soil, a Fluvisol soil, and AA soil, a Plinthosol soil. Basic information on the soils collected for the greenhouse experiment is presented in Fig. 1(A). The RA soil was from Binh Tan district, Vinh Long Province, Vietnam, with latitude and longitude coordinates of 10.118756, 105.733604; (B) the AA soil was from Tan Hong District, Dong Thap Province, Vietnam, with latitude and longitude coordinates of 10.900970, 105.433574.

### Climatic conditions

From 2020 to 2022, the average monthly temperature in the MD was 28 °C, with the highest temperature (29 °C) recorded in May and the lowest temperature (26 °C) recorded in January. During 2019–2021, the average monthly rainfall was 175 mm; the highest rainfall

(A)                                                  (B)

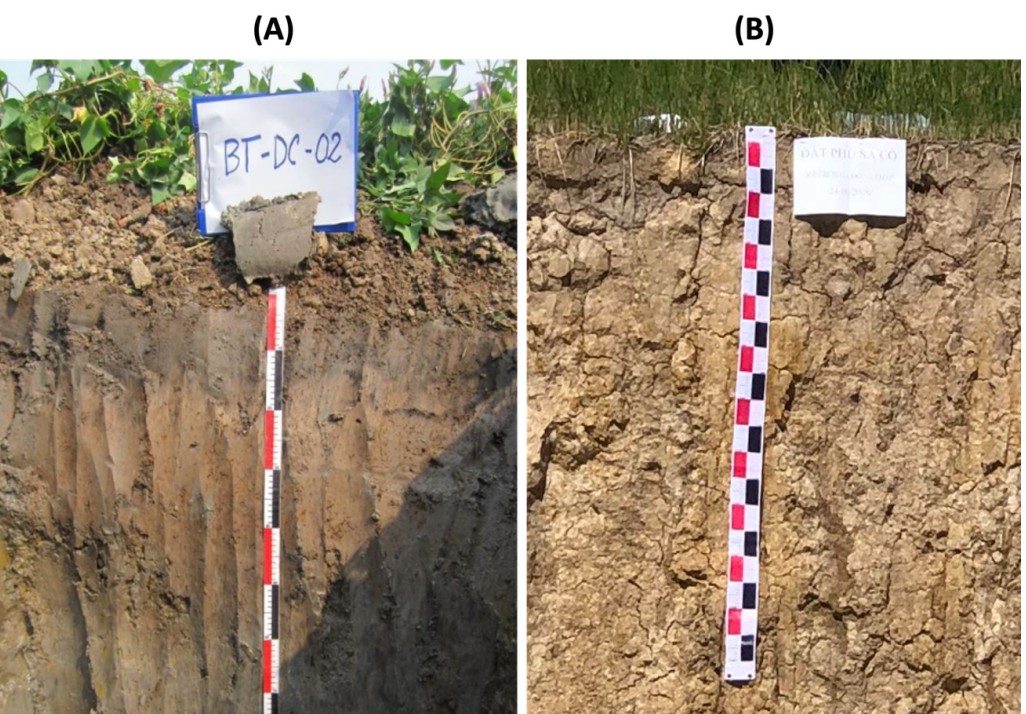

**Figure 1  Soil profiles of (A) recent alluvial soil with sweet potato vegetation; (B) ancient alluvial soil with wet rice vegetation.** Photos were taken in April 2021.

(300 mm) occurred in September, and the lowest (10 mm) in January. The meteorological data for both locations in the MD region are shown in Fig. 2.

## Varieties of blueberry

The blueberry varieties used for pot experiments were obtained from the Center of Agricultural Crops, Vietnam National University of Agriculture (*Center of Agricultural Crops, 2023*). Three-year-old varieties of blueberry from the micropropagation project of Can Tho University were used in the greenhouse experiment. The genus of the blueberries was identified using DNA extracted from young leaves (*Rogers & Bendich, 1989*). DNA was extracted at the Laboratory of Molecular Biology, Can Tho University, Vietnam. The Basic Local Alignment Search Tool was used, and the internal transcribed spacer (ITS) gene sequences of the blueberry samples were compared with the gene sequences of other species in the GenBank database of the National Center for Biotechnology Information. The results showed that the ITS gene sequence of the blueberry sample was similar to that of AF273709.1 (*Floyd, 2002*), as determined by Quynh (96.84%; *Quynh et al., 2023*).

## Chemical characteristics of the compost

The compost used in this study was a commercial product of PPE Co., Ltd. (Can Tho, Vietnam). Chemical characteristics of the compost were: pH, 8.70; total C, 154 g kg$^{-1}$; soluble Na, 1.59 cmol$_c$ kg$^{-1}$; soluble K, 20.0 cmol$_c$ kg$^{-1}$; soluble Ca, 7.29 cmol$_c$ kg$^{-1}$, total

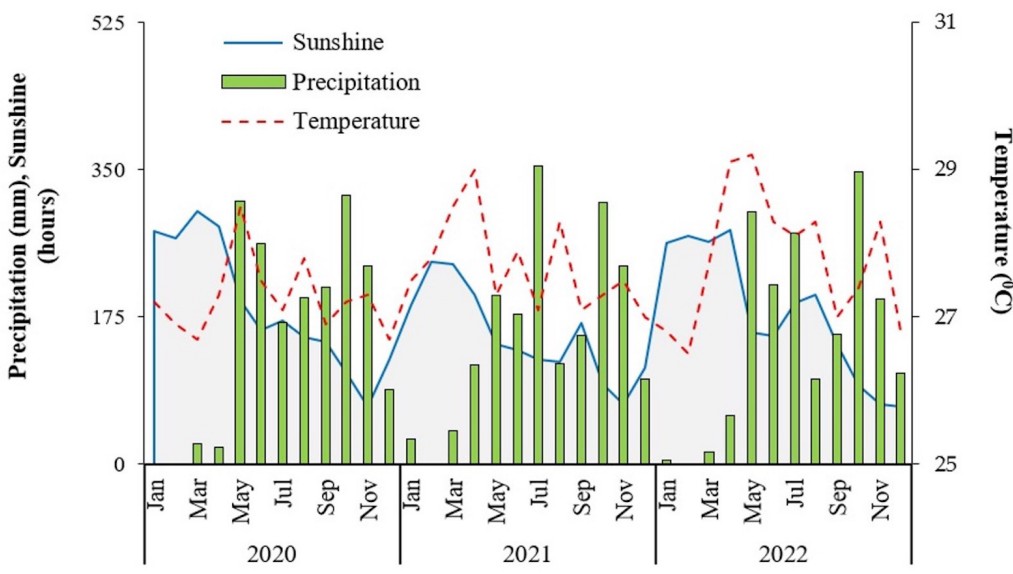

**Figure 2** Climatic conditions during 2020 and 2022 in Can Tho City, Vietnam (Source: Station Meteorology and Hydrology of Can Tho City, Vietnam).

**Table 1** Two-factor factorial design including soil types and omission of fertilizer application to the blueberry plants.

| Factor levels | Soil types | |
| --- | --- | --- |
| Omission fertilizer application | RA | AA |
| NPK | TRT–1 | TRT–5 |
| PK | TRT–2 | TRT–6 |
| NK | TRT–3 | TRT–7 |
| NP | TRT–4 | TRT–8 |

Notes.

RA, Recent alluvial soil; AA, Ancient alluvial soil.

porosity, 76.2%; exchangeable Na, 0.60 $cmol_c$ $kg^{-1}$; exchangeable K, 15.0 $cmol_c$ $kg^{-1}$, and exchangeable Ca, 61.6 $cmol_c$ $kg^{-1}$.

## Experimental design

The experiment was performed in the greenhouse of Can Tho University beginning in June 2021 with a factorial randomized block design (RBD) based on two factors: (a) two soil types (RA and AA) and (b) four omission treatment conditions (NPK, PK, NK, and NP; Table 1). Each treatment was replicated four times. The distance between the pots was 30 cm × 30 cm.

The following fertilizer formula for blueberries was used, as described by *Marty et al. (2019)*: 45 kg N $ha^{-1}$, 20 kg $ha^{-1}$ $P_2O_5$, and 20 kg $ha^{-1}$ $K_2O$. The fertilizers used were ammonium sulfate (21% N), triple superphosphate (45% $P_2O_5$), and potassium sulfate (52% $K_2O$). Ammonium sulfate is used in blueberries because its ammonia form is absorbed

by plants more efficiently than nitrate (*Percival & Privé, 2002*; *Merhaut & Darnell, 1995*; *Erick & James, 2019*).

As recommended by *Rutgers University (2012)*, the application of N fertilizer was split into three time points: 3–4 weeks (April 10, 2022), 7–8 weeks (May 10, 2022), and 11–12 weeks (June 10, 2022). P and K fertilizers were applied in March, before bud break (*Hart et al., 2006*).

The pots used were 14 cm × 12 cm × 20 cm in size, with 10 kg of soil added per pot, and 50 g of OM was applied to each pot. Before planting the blueberries, the soil in the plant root zone was mixed with compost, with a root zone size of approximately 10 cm ×5 cm/2 L.

## Soil sampling and analysis

The soils for the pot experiment were taken from Hau Giang (10° 7′7.521″ N, 105° 44′0.9738″ E) and Dong Thap (10° 54′3.4914″ N, 105° 26′0.8658″ E). A soil auger (3 cm in diameter) was used to collect five soil cores from depths of 0–20 and 20–40 cm. At the same depth, the soil was subsampled at approximately 500 g and each soil depth was blended into one composite sample. All soil samples were air-dried, crushed, and sieved through a two mm sieve.

The chemical properties of soil samples were then determined. Soil pH (soil:water suspension of 1:2.5) was determined using a digital pH meter (Metrohm 744). Soil organic carbon was determined using the Walkley and Black method (*Walkley & Black, 1934*). The total soil N was determined using the semi-micro-Kjeldahl method after digesting the samples with $H_2SO_4$ (*Bremner, 1965*). Ammonium phosphomolybdate and the Bray-2 method were used to determine the total and available P levels, respectively (*Bray & Kurtz, 1945*).

The cation exchange capacity (CEC) and exchange cations ($K^+$, $Ca^{2+}$, and $Mg^{2+}$) of the soils were determined using the 0.1 M BaCl2-compulsive exchange procedure (*Gillman & Sumpter, 1986*; *Rhoades, 1982*). CEC was determined by titrating the extractant with barium chloride-triethanolamine, which was buffered at pH 8.2 (*Mehlich, 1938*). The exchange cation extractants ($K^+$, $Ca^{2+}$, and $Mg^{2+}$) were determined using flame photometry (*Houba, Vanderlee & Novozamsky, 1995*).

## Sampling and analysis of stems, leaves, and fruits

At the time of harvest, the low bush blueberries were separated into stems, leaves, and fruits, and the fresh stem and leaf tissues were oven-dried at 70 °C for three days to achieve a constant weight. The dry weight was recorded and the dried tissues were ground in a pulverizer and sieved through a two mm sieve. A freeze-drying method was used to obtain dried samples of fresh blueberry fruits. Twenty fresh fruits (about 5g) were cut in half lengthwise, then placed in special freeze-drying bottles and frozen at −40 °C for one hour, followed by immediate freeze-drying for 72 h in an ultra-freezer at −40 °C. A Labconco FreeZone 4.5 Liter Benchtop Freeze Dry System (7750020) was used at a pressure of 0.002 mbar. After freeze-drying, the fruit was placed in a vacuum-sealed plastic bag to analyze its mineral composition (N, P, and K).

The N, P, and K content of the plant parts were determined at the laboratory of the Soil Science Department, College of Agriculture, Can Tho University. The dried samples were digested in a mixture of $H_2SO_4$ and $H_2O_2$, and then the N concentration of the air-dried samples was determined using the Kjeldahl method. The P concentration was determined by the intensity of the blue color of molybdenum produced by the reaction of ammonium molybdate with ascorbic acid using a UV spectrophotometer (*Jackson, 1958*), and the K concentration was determined photometrically using an air-acetylene flame.

## Plant growth and biomass

At the time of fruit harvest, the number of leaves and stems on the plant were counted and plant height was measured. The number of plants in each treatment was measured using four replicates. To determine leaf and stem dry weights, the leaves and stems were dried in an oven for 24 h.

Since the blueberries in the shrub did not ripen simultaneously, fruit yield was determined by the cumulative number of fruits on the treatment tree after each of the four harvests. At each harvest, all fruits on the tree were weighed. The mean fruit weight was determined based on the total weight (kg) of fruits per plant.

## Determination of total N, P, and K uptake and nutrient recovery efficiency

Estimates of indigenous soil nutrient supply capacity, such as soil N, are based on the principle of "essential elements" in soils where plants that successfully grow in soil that is not fertilized with N, that must be received N from the soil. The omission plot technique is useful for estimating the indigenous soil nutrient supply capacity and plant uptake of fertilizer (*Dobermann, 2007*). In this study, N, P, and K were applied to the soil in adequate amounts to ensure that the nutrients were not limited by an insufficient supply from the soil. The difference in N uptake between a plot with adequate fertilizer application and a plot with N omission was considered the gap between the amount of N needed by the crop and the indigenous soil N supply; this gap determined the total amount of N fertilizer needed.

## Estimation of total N, P, and K uptake

Total N uptake was calculated by multiplying the dry weight by the N concentration in the grain, straw, and fruit. Total P and K uptake were calculated the same way.

## Estimation of soil indigenous nutrient supply

Plant N accumulation in the aboveground dry matter at harvest in the N omission plot (PK) was considered the indigenous N supply (INS), and plant K and P accumulation in the aboveground dry matter at harvest in the K and P omission plots (NK and NP) were considered the indigenous P and K supply (IPS and IKS), respectively.

## Apparent nutrient recovery efficiency

Apparent nutrient recovery efficiency (ARE) is a measure of the crop's ability to absorb nutrients from the soil. It specifically measures the proportion of the applied nutrient that

**Table 2** Physiochemical characteristics of the two topsoil samples (0–20 cm) collected for the greenhouse experiment.

| Indicator | unit | Soil type | |
|---|---|---|---|
| | | RA | AA |
| $pH_{H2O}$ | | 4.96 | 4.6 |
| EC | $mS\ cm^{-1}$ | 0.31 | 0.12 |
| Availble P (Bray-2) | $mg\ kg^{-1}$ | 9.1 | 10.9 |
| Organic matter | % | 7.84 | 1.38 |
| CEC | $cmol_c\ kg^{-1}$ | 15.8 | 5.35 |
| *Exchangeable cation* | | | |
| $K^+$ | | 0.18 | 0.17 |
| $Ca^{2+}$ | $cmol_c\ kg^{-1}$ | 7.36 | 2.12 |
| $Mg^{2+}$ | | 3.66 | 0.45 |
| Bulk density | $g\ cm^{-3}$ | 1.03 | 1.31 |
| *Particle size* | | | |
| Sand | | 0.59 | 19.8 |
| Silt | % | 55.02 | 65.4 |
| Clay | | 44.4 | 14.8 |
| Soil texture classification | | Silty clay | Silty loam |

**Notes.**
RA, Recent alluvial soil; AA, Ancient alluvial soil.

is taken up by the aboveground plant biomass (*Dobermann, 2007*), calculated as follows:

$$\text{ARE for N, P or K (\%)} = \frac{U - U_0 \times 100}{F} \qquad (1)$$

where U is the amount of N, P, or K taken up by the stem, leaves, and fruit in the NPK treatment, $U_0$ is the amount of N, P, or K taken up by the stem, leaves, and fruit in the omission treatments, and F is the amount of N, $P_2O_5$, or $K_2O$ fertilizer applied ($kg\ ha^{-1}$).

## Statistical analysis

The mean and standard deviation ($\pm$ SE) was calculated for each treatment. The SPSS software (version 20.0) was used to conduct a two-way analysis of variance (ANOVA), and Duncan's multiple range test was used for multiple comparisons with significance levels of 5% and 1%. The relationships among plant dry weight, nutrient (N, P, and K) content, and nutrient uptake were determined using Pearson's correlation coefficient.

## RESULTS

### Characteristics of RA and AA soils

The pH values of RA and AA soils are both less than 5.0 (Table 2) because in both soil profiles, pyrite material appears at a depth of more than 50 cm (*Ngoc et al., 2023*; *Khoa et al., 2023*), and under aerated conditions, acidity occurs due to the oxidation of pyrite.

RA and AA (Table 3) soils are both formed by the Mekong sediments, but with different accretion times. AA soil (150,000 ha) is found in the northeast region of the delta, along the Cambodia-Vietnam border, with deposition starting 11,000 years ago (*Chiem, 1993*).

**Table 3  Effect of NPK omission and soil types on NPK content in blueberry plant parts. Greenhouse experiment was conducted at the College of Agriculture, Can Tho University, May 2022.**

| Treatments | Stem mineral content (g kg$^{-1}$) | | | Leaf mineral content (g kg$^{-1}$) | | | Fruit mineral content (g kg$^{-1}$) | | |
|---|---|---|---|---|---|---|---|---|---|
| | N | P | K | N | P | K | N | P | K |
| **Omission** | | | | | | | | | |
| NPK | 6.45a | 3.45a | 7.58a | 19.12a | 1.39a | 5.71a | 11.02a | 1.09a | 5.88a |
| PK | 4.83c | 2.63b | 7.07b | 10.42d | 0.84d | 3.79c | 8.48b | 0.61c | 3.34d |
| NK | 5.7b | 2.67b | 7.17ab | 14.55c | 1.03c | 4.95b | 8.88b | 0.73bc | 4.53b |
| NP | 5.99b | 3.30a | 6.26c | 16.56b | 1.28b | 4.91b | 10.48a | 0.82b | 4.08c |
| **Soil types** | | | | | | | | | |
| RA | 6.29 | 3.00 | 7.30 | 16.75 | 1.20 | 5.33 | 10.60 | 0.91 | 5.10 |
| AA | 5.02 | 3.03 | 6.74 | 13.59 | 1.08 | 4.35 | 8.83 | 0.71 | 3.82 |
| F (A) | ** | ** | ** | ** | ** | ** | ** | ** | ** |
| F (B) | ** | ns | ** | ** | ** | ** | ** | ** | ** |
| F (AxB) | ** | * | ns | ** | ns | * | * | * | ns |
| CV(%) | 15.6 | 15.5 | 9.8 | 24.9 | 21.4 | 20.1 | 16.2 | 30.1 | 27.6 |

Notes.

RA, Recent alluvial soil; AA, Ancient alluvial soil.

[*,**]Significant at the 0.05 and 0.01 probability levels, respectively. Values in the same column with the same lowercase letter are not significantly different ($P > 0.05$).

RA soil (3,900,000 ha) constitutes about 70 percent of the MD. It was formed by sediment deposition in the last 6,000 years and is found at the southern tip of Vietnam (*Sub-Institute of Hydrometeorology and Environment of South Vietnam, 2010*).

The exchangeable values (cmol$_c$ kg$^{-1}$) of Ca$^{2+}$ (7.36), Mg$^{2+}$ (3.66), CEC (15.38), and content of OM (7.84%) in RA soil were much higher than those in the AA soil (Ca$^{2+}$ (2.12), Mg$^{2+}$ (0.45), CEC (5.35), and OM (1.38%). As a result, because of AA's low cation absorption capacity, the nutrient content of AA soil was approximately a quarter of the nutrient content of RA soil.

## Plant dry weight

Images of blueberry growth in the two-factor experiment are presented in Fig. 3. The RA soil type consisted of N, P, and K omission treatments arranged in rows in four replicates. In the first row, when comparing PK, NK and NP with NPK, the results show that plants in the PK treatment (without N fertilization) have lower height, number of branches, flowers and leaves compared to NPK, while NK and NP is less different in terms of crop growth other than NPK (Fig. 3A). For AA soil, plant growth responses to N, P and K omission are similar to RA soil, that is, plants in PK also have lower height, number of branches, flowers and leaves compared to those of NPK (Fig. 3B).

Comparison of plant growth responses under RA and AA soils is shown in Fig. 4. Plant growth in RA soil has better height, number of branches, flowers and leaves than that of AA soil.

In relation to the plant growth, the biomass of different plant parts (stem, leaves, and fruit) in both RA and AA soils, was consistently higher in the NPK treatment than in the PK, NK, and NP treatments, and the biomass of the different plant parts was the lowest in the omission of N (PK) treatment (Fig. 5). In RA soil, the fruit weight (g plant$^{-1}$) of

**(A)**

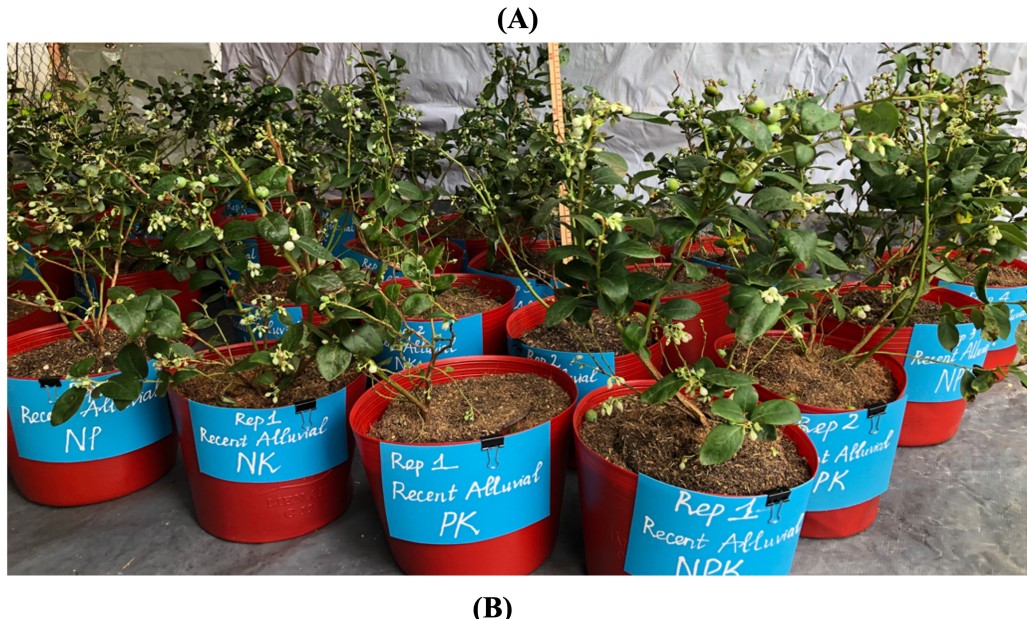

**(B)**

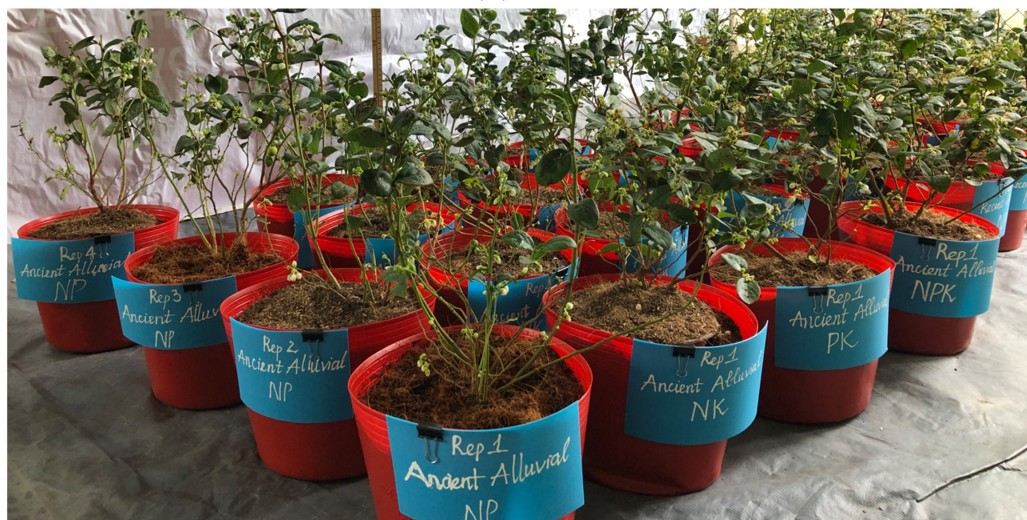

**Figure 3** **Effect of N, P and K omission on blueberry plant growth in (A) recent alluvial soil and (B) ancient alluvial soil.** College of Agriculture, Can Tho University. April 2022.

NPK, PK, NK, and NP (10.5, 4.94, 7.00, and 8.15, respectively) varied according to the leaf biomass (g plant$^{-1}$; 5.92 in NPK, 6.48 in PK, 4.44 in NK, and 4.98 in NP). Similarly, in AA soil, the fruit weights (g plant$^{-1}$) of NPK, PK, NK, and NP (8.50, 3.30, 5.65, and 5.63, respectively) varied according to leaf biomass (g plant$^{-1}$; 5.45 in NPK, 2.49 in PK, 3.49 in NK, and 3.87 in NP). By using the same fertilizer formula, 45N–20P$_2$O$_5$–20K$_2$O mixed with CHC (10 tha$^{-1}$) in the pot soil, growth during the fruit production stage of blueberries indicated that the fertility of RA soil was higher than that of AA soil. When comparing fruit yield between RA soil (10.5 g plant$^{-1}$) and AA soil (8.50 g plant$^{-1}$), blueberry yield in AA soil was only 81% of blueberry yield in RA soil (Fig. 5). In this study, the application of
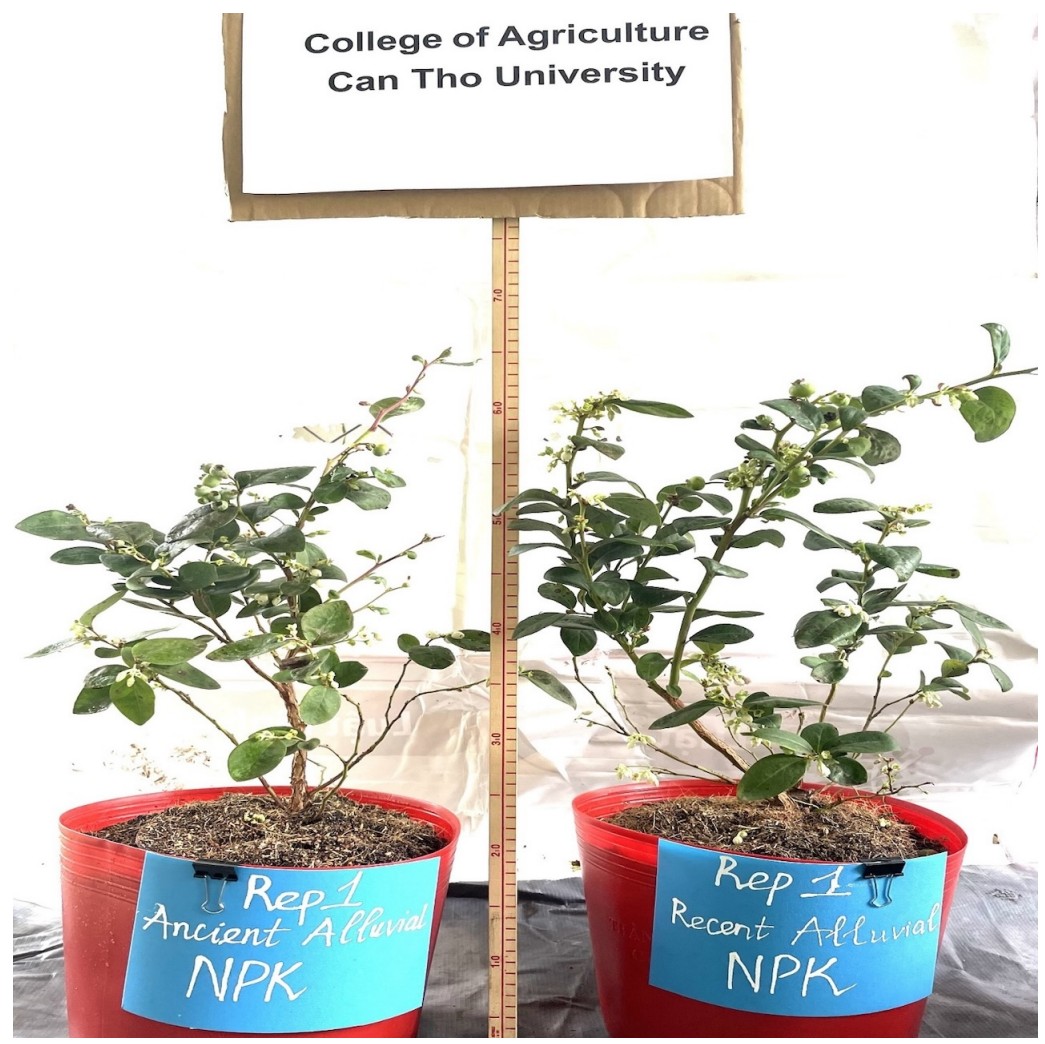

**Figure 4  Effect of of N, P and K omission on blueberry plant growth in AA and RA soil.** College of Agriculture, Can Tho University. April 2022.

45 kg N ha$^{-1}$ was based on the recommendation of *Marty et al. (2019)*. Using ammonium sulfate at this concentration is considered highly suitable for blueberry plants because it quickly transforms to $NH_4$-N and helps acidify the soil, which significantly decreases $NH_3$ volatilization (*Powlson & Dawson, 2021*). However, the accumulation of sulfate in this fertilizer can be harmful to plants. Therefore, it is necessary to use ammonium sulfate at recommended levels (*Messiga, Haak & Dorais, 2018*).

### Effect of N, P, and K omission on nutrient content

Table 3 shows that N fertilizer was the most important factor affecting N, P, and K content in blueberry stems, leaves, and fruits. Leaf N, P, and K (g/kg) contents in the PK treatment group (10.42, 0.84, and 3.79, respectively) were significantly lower than those in the NPK treatment (19.12, 1.39, and 5.71, respectively) group.

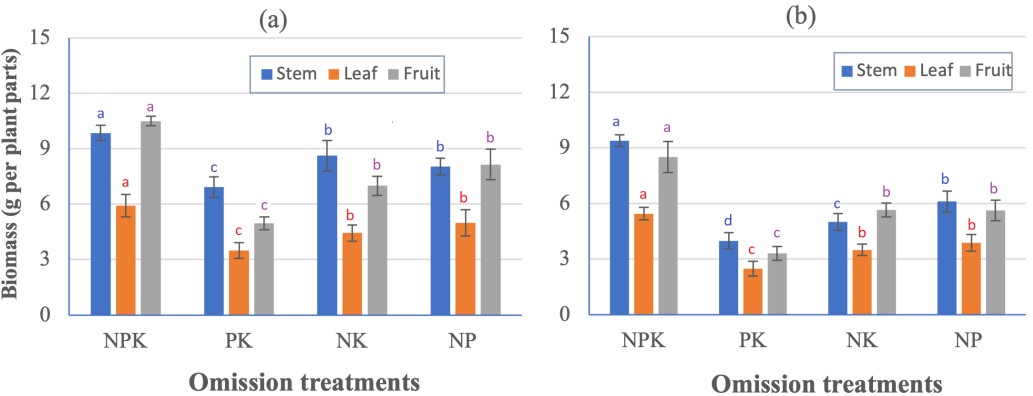

**Figure 5  Effect of N, P and K omission on blueberry plant biomass (g per plant parts) in two soils.** (A) Recent alluvial soil, and (B) ancient alluvial soil. Values with the same lowercase letter above bars in columns of the same color are not significant.

According to *Trevett (1972)*, the sufficiency ranges of N, P, and K in the leaves of lowbush blueberries are 16.0–23.8, 1.2–2.2, and 4.0–9.0 g kg$^{-1}$, respectively. The leaf N, P, and K contents following N omission (PK) were lower than the thresholds for leaf tissue N, P, and K established by *Trevett (1972)*. The N, P, and K contents in the stems, leaves, and fruits after the omission of P (NK) and K (NP) were consistently lower than those in the stems, leaves, and fruits of plants with adequate NPK fertilizer application. Moreover, after P omission (NK), leaf P content (1.03 mg kg$^{-1}$) was below the threshold for P posited by *Trevett (1972)*.

The N, P, and K contents in the stems, leaves, and fruits of blueberries in RA were almost all significantly higher than those in the stems, leaves, and fruits of plants grown in AA (Table 3). An interaction between treatment (A) and soil type (B) was found for N in the stems (**), leaves (**), and fruits (*).

### Effects of nutrient omission on N, P, and K uptake

Among the four treatment conditions—NPK, PK, NK, and NP—the total uptake of N, P, and K was the highest in the NPK treatment group (45.5, 9.1, and 29.9 mg plant$^{-1}$, respectively) and lowest in the PK treatment group (20.5, 4.1 and 13.3 mg plant$^{-1}$; Fig. 6). The low uptake of N, P, and K in the N omission treatment group (PK) occurred in parallel with the N, P, and K contents in the leaves (Table 4). With the omission of P and K, the N, P, and K uptake by the blueberry plants was also reduced, but the decrease was not as severe as that observed in the PK treatment group.

### Correlation matrix for blueberry biomass and mineral nutrients

The interrelationships among the plant parameters were described using a correlation matrix (Fig. 7). There was a positive correlation between the dry weight of the plant parts; Pearson's r of the stem, leaf, and fruit weight ranged from 0.86–0.93. Moreover, the

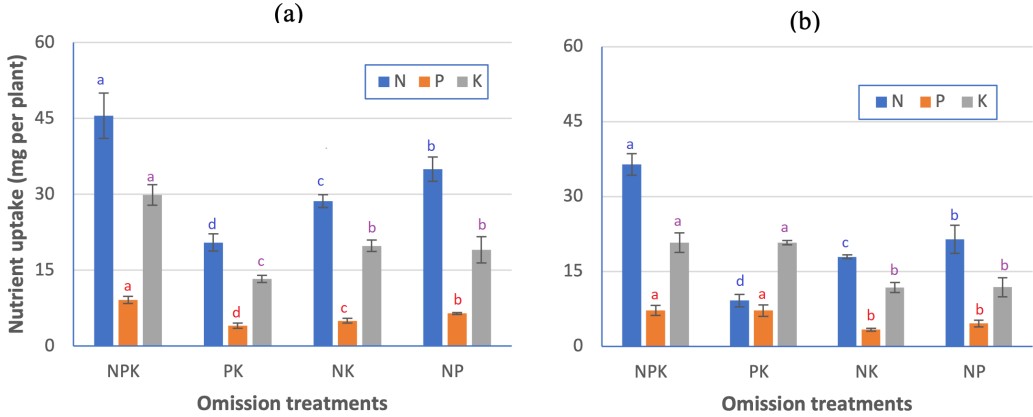

**Figure 6  Effect of N, P, and K omission on the uptake of N, P, and K in plant parts of blueberries grown in (A) recent alluvial soil, and (B) ancient alluvial soil.** Values with the same lowercase letter above bars in columns of the same color are not significant.

**Table 4  Nutrient uptake after omission treatments and apparent nutrient recovery efficiency in blueberry grown in the two soil types.** The total content of phosphorus and potassium are expressed as $P_2O_5$ and $K_2O$, respectively.

| Nutrient uptake by plant | Mineral NPK uptake (kg ha$^{-1}$) | | | | | |
|---|---|---|---|---|---|---|
| | RA | | | AA | | |
| | N | $P_2O_5$ | $K_2O$ | N | $P_2O_5$ | $K_2O$ |
| Nutrient taken up in soil and fertilizer by plant (NPK treatment) | 45.5 | 9.14 | 29.9 | 36.4 | 7.20 | 20.8 |
| INS (PK treatment) | 20.5 | | | 9.2 | | |
| IPS (NK treatment) | | 5.03 | | | 3.36 | |
| IKS (NP treatment) | | | 19.0 | | | 11.9 |
| N uptake from N fertilizer ($U_{NPK}$-$U_{PK}$) | 25.1 | | | 27.2 | | |
| P uptake from N fertilizer ($U_{NPK}$-$UN_{NK}$) | | 4.11 | | | 3.84 | |
| K uptake from K fertilizer ($U_{NPK}$-$U_{NP}$) | | | 10.9 | | | 8.9 |
| *ARE (%)* | *55.7* | *20.6* | *54.3* | *60.6* | *19.2* | *44.7* |

**Notes.**
*Using fertilizer formula $45N–20P_2O_5–20K_2O$.
*Italic numbers* represent standard error (SD). RA, Recent alluvial soil; AA, Ancient alluvial soil.

correlations between the fruit uptake of N and P ($r = 0.95$), P and K ($r = 0.91$), and K and N ($r = 0.96$) were strong.

There were significant correlations between N, P, and K uptake and dry weight (Fig. 7). The positive correlation between N uptake and the weights of stems, leaves, and fruits ($r = 0.92$, 0.97, and 0.97, respectively) was higher than the strong correlations observed between P uptake and the weights of stems, leaves, and fruits ($r = 0.88$, 0.92, and 0.91, respectively) and between K uptake and the weights of stems, fruits, and leaves ($r = 0.91$, 0.92, and 0.95, respectively).

| Stem wt | 0.88 | 0.86 | 0.84 | 0.43 | 0.40 | 0.89 | 0.71 | 0.80 | 0.78 | 0.68 | 0.78 | 0.92 | 0.88 | 0.91 |
|---|---|---|---|---|---|---|---|---|---|---|---|---|---|---|
| | Leaf wt | 0.93 | 0.85 | 0.60 | 0.36 | 0.91 | 0.78 | 0.81 | 0.75 | 0.74 | 0.80 | 0.97 | 0.92 | 0.92 |
| | | Fruit wt | 0.82 | 0.61 | 0.43 | 0.90 | 0.83 | 0.83 | 0.74 | 0.78 | 0.86 | 0.97 | 0.91 | 0.95 |
| | | | Stem N | 0.47 | 0.28 | 0.91 | 0.73 | 0.74 | 0.86 | 0.69 | 0.78 | 0.88 | 0.77 | 0.83 |
| | | | | Stem P | 0.03 | 0.59 | 0.64 | 0.29 | 0.60 | 0.49 | 0.37 | 0.62 | 0.68 | 0.47 |
| | | | | | Stem K | 0.22 | 0.05 | 0.42 | 0.25 | 0.40 | 0.61 | 0.41 | 0.35 | 0.58 |
| | | | | | | Leaf N | 0.85 | 0.80 | 0.87 | 0.68 | 0.77 | 0.94 | 0.84 | 0.85 |
| | | | | | | | Leaf P | 0.74 | 0.74 | 0.81 | 0.73 | 0.84 | 0.83 | 0.76 |
| | | | | | | | | Leaf K | 0.67 | 0.70 | 0.80 | 0.84 | 0.73 | 0.87 |
| | | | | | | | | | Fruit N | 0.64 | 0.68 | 0.83 | 0.74 | 0.72 |
| | | | | | | | | | | Fruit P | 0.85 | 0.80 | 0.84 | 0.83 |
| | | | | | | | | | | | Fruit K | 0.86 | 0.80 | 0.93 |
| | | | | | | | | | | | | N Uptake | 0.95 | 0.96 |
| | | | | | | | | | | | | | P Uptake | 0.91 |
| | | | | | | | | | | | | | | K Uptake |

**Figure 7** Correlation matrix among plant weights, nutrient content, and uptake of N, P, and K in blueberry.

## Nutrient uptake and apparent nutrient recovery

After NPK treatment of the RA soil (Table 4), total N uptake was 45 kg N ha$^{-1}$, and the INS (PK treatment) was 20.5 kgN ha$^{-1}$. N uptake from N fertilizer (25.1 kgN ha$^{-1}$) was calculated as the difference in N uptake between the non-omission and omission of N treatments (U$_{NPK}$-U$_{PK}$), and P and K uptake were calculated the same way. Mineral NPK uptake in AA soils was interpreted similarly for RA soil.

ARE is the main indicator of the nutrient uptake characteristics of plants. The removal of nutrients from harvested crops was estimated using Formula 1, which is commonly used to explain nutrient efficiency. Table 4 shows that in the RA soil, the ARE of N (55.7%) was lower than that in the AA soil (60.6%), whereas the ARE of P$_2$O$_5$ did not differ between the two soils. The ARE of K$_2$O in RA soil (54.3%) was much higher than that in AA soil (44.7%).

## DISCUSSION

### Soil characteristics of RA and AA soil in the MD

*CEC* is a useful indicator of soil fertility. Soil CEC mainly depends on the amount of organic matter and type of clay minerals found in the soil. The properties of alluvial soils are greatly dependent on the predominant clay minerals (*Newman, 1984*). Among the different clay minerals, kaolinite has the lowest CEC (approximately 10 cmol kg$^{-1}$; *Thanh & Egashira, 2000*) whereas the CEC of illite/smectite and OM are higher (250–400 and 25–100 cmol kg$^{-1}$, respectively; *Brown & Lemon, 2016*). Since the RA soil of the MD has a heavy texture and is rich in kaolinite (55%; *Chiem, 1993*) with low OM (1.38%), the CEC of AA soil (5.35 cmol kg$^{-1}$) is much lower than that of RE soil (15.8 cmol kg$^{-1}$; Table 3).
## Plant dry weight, nutrient content, and plant uptake

Compared to P and K omission in RA and AA soils, N omission resulted in the lowest plant biomass (stems, leaves, and fruits; Fig. 5). N deficiency has the greatest impact on plant growth, fruit development, and flower bud formation and differentiation (*Percival & Sanderson, 2004*). However, even with N omission, blueberries can still grow using the N stored in the plant from the previous season (*Loescher, McCamant & Keller, 1990*). Furthermore, the N, P, and K provided by OM fertilizer application to the soil at the beginning of the pot experiment may have partially supplied nutrients during omission treatments. Organic matter improves soil fertility by providing nutrients through mineralization, improving water-holding capacity, creating roots, and aerating the living environment for soil microorganisms (*Fischer & Glaser, 2012*).

In the N omission treatment, the concentrations of N (10.42 g N kg$^{-1}$), P (0.84 g P kg$^{-1}$), and K (3.79 g K kg$^{-1}$) in the blueberry leaves (Table 4) were below the Trevett threshold. Under the omission treatments, plant uptake varied according to the reduction in biomass and nutrient content, with N omission leading to the lowest N, P, and K uptake (Fig. 6).

Most previous studies apply the Trevett threshold to Vaccinium angustifolium, but no specific threshold range has been established for evaluating leaf nutrient content for Vaccinium tellenum. Some previous studies have used Trevett's range for evaluating other blueberry cultivars, such as highbush blueberry Vaccinium corymbosum L (*Szwonek, 2004*) and half-highbush 'Aino,' 'Alvar,' 'Arne,' and 'Northblue' blueberries (*Tasa et al., 2012*).

By using the Trevett threshold in this study, in this study, the N and P concentration of blueberry leaves fell to the lowest level of the Trevett threshold with N and P omission treatment, while the N and P content in the leaves of blueberry plants fertilized with N and P both reached the normal Trevett threshold

In the present study, N deficiency caused low vegetative growth and reproduction, P deficiency reduced energy transfer in plants and roots (*Florida Blueberry Growers Association, 2023*), and K deficiency reduced photosynthesis, metabolism, and subsequent carbohydrate metabolism (*Pettigrew, 2008*; *Zörb, Senbayram & Peiter, 2014*; *Lu et al., 2016*). Fertilizer application is usually not recommended based on soil nutrient levels because these indicators are not always the same as those for leaves; for example, sometimes the P content in the soil is low, but P present in the leaves is not low. Therefore, fertilizer recommendations for lowbush blueberries are primarily based on leaf tissue tests; however, soil testing complements the leaf tissue tests (*Yarborough & Smagula, 2013*).

Several studies have shown that the amount of available N in the soil and the N status of plants greatly affect the uptake of P and K. This study verified that N deficiency leads to low P and K content (Table 3). Increased crop yields are mainly related to N fertilizer (*Maqbool et al., 2016*), so N fertilizer application is important for enhancing fruit yield by increasing the N content in the leaves (*Reddy et al., 2001*).

## Correlation matrix between blueberry biomass and mineral nutrients

Figure 7 shows a positive correlation between N, P, and K contents in stems, leaves, and fruits (Pearson's *r* range of 0.67–0.85 in stem, 0.73–0.88 in leaf, and 0.68–0.71 in fruit).

According to *Kamprath (1987)*, there is a positive correlation between the content of leaf P and fruit P and the accumulation between total P and total N. Results from traditional and organic farming studies have shown that the N, P, and K contents in leaves are all positively correlated (*Vilhena et al., 2022*), reflecting the dilution of these elements, followed by an increase in leaf and stem biomass. Leaves with low P and K content also have low N content. These results indicate that an increase in crop yield is mainly associated with N fertilizer application (Parent, 2013; *Maqbool et al., 2016*). Therefore, according to *Terry (2008)*, balanced fertilizer application of N, P, and K results in better yields and plant uptake because the interaction between N, P, and K improves crop yield and N use efficiency.

In this study, the N, P, and K contents in leaves were positively correlated with fruit yield, which agrees with the findings of *Quesnel et al. (2006)* that the most important factor is soil OM and there is a positive correlation between the nutrient content in leaves and soil OM. These results are also consistent with those reported by *Trevett, Carpenter & Durgin (1968)*.

## Nutrient uptake and apparent nutrient recovery

Four agronomic indices are commonly used to express nutrient use efficiency (NUE): partial factor productivity, agronomic efficiency, physiological efficiency, and apparent recovery efficiency (*Mosier, Syers & Freney, 2004*). Crop removal efficiency is also commonly used to explain NUE (*Congreves et al., 2021*).

NUE indices are related to grain yield, soil indigenous nutrient supply, fertilizer application rates, and integrated crop management practices (*Dobermann, 2007*). Therefore, the attainable yield and indigenous soil nutrient supply are the most important data for providing a scientific basis for nutrient management. However, because the NUE estimation ignores the soil N supply, it can produce inconsistent results due to variability in the soil inorganic N supply (*Congreves et al., 2021*). Moreover, with the NUE estimation, some assumptions about the N supply from soil sources such as N fertilizer do not affect soil mineralization, soil fixation, or N loss from omission plots (*Huggins & Pan, 1993*).

A comparative evaluation of INS between the RA and AA soils showed that RA soil had higher soil N uptake (20.5 kg ha$^{-1}$) compared to the AA soil (9.2 kg ha$^{-1}$; Table 4), but ARE showed the opposite trend in both soil types. The ARE (55.7%) of RA soil was lower than the ARE (60.6%) of AA soil. INS has a less positive impact on the ARE index when the soil INS is high. This study concurs with several recent publications that show that ARE has a positive response when the soil has low INS, but ARE has a less positive response to N fertilizer when soils have high INS (*Espe et al., 2015*; *Awio et al., 2023*). Research on rice soil has also shown that the ARE of N, P, and K decreases as INS increases (*Haefele et al., 2003*; *Haefele & Wopereis, 2005*).

Table 4 shows that the ARE indexes of K in RA and AA (54.3 and 44.7%, respectively) are higher than those of P in RA and AA (20.6 and 19.2%, respectively). K is less vulnerable to losses than P, although K is very soluble and therefore susceptible to leaching or runoff. Soil K loss is negligible because it can be adsorbed by clay minerals and organic materials (*Goulding et al., 2021*).

Crop yields can be improved with fertilizer inputs to better meet crop requirements. Economic efficiency is achieved when balancing plant yield with costs of providing sufficient nutrients for plants, but finding this balance is difficult because these factors are not known in advance of the growing season (*Terry, 2008*).

## CONCLUSIONS

RA and AA soil types both have a high clay content, but differ in soil-supplied mineral composition and respond differently to crop productivity. In using the same fertilizer formula, $45N–20P_2O_5–20K_2O$, and mixing CHC (10 tha$^{-1}$) into pot soil, the blueberry yield in AA soil only reached 81% of the blueberry yield in RA soil.

In both RA and AA soils, N omission caused leaf and foliar N deficiencies, resulting in foliar P and K content levels falling below the Trevett threshold. N omission resulted in reduced stem and leaf weights and reduced fruit yield. Therefore, the increased crop yield in both soils was mainly related to N fertilizer application.

The INS of RA, the fertile soil, was high, but the ARE index was low, while the INS of AA, the degraded soil, was lower but yielded a higher ARE index. The higher the INS level, the less positive the impact on the ARE index. Therefore, in AA soils, the indigenous N and K supply can be improved through fertilizer investment, but a balance must be achieved with economic efficiency.

## ACKNOWLEDGEMENTS

We thank Ms. Huynh Ngoc Truyen and graduate students Le Thi Sen and Nguyen Thanh Tung for helping with the agronomic parameter measurements.

### Funding
This work was supported by the Ministry of Education and Training, Vietnam, with grant number: B2021-TCT-10. The funders had no role in study design, data collection and analysis, decision to publish, or preparation of the manuscript.

### Grant Disclosures
The following grant information was disclosed by the authors:
The Ministry of Education and Training, Vietnam: B2021-TCT-10.

### Competing Interests
The authors declare there are no competing interests.

### Author Contributions
- Ngo Phuong Ngoc conceived and designed the experiments, performed the experiments, analyzed the data, prepared figures and/or tables, authored or reviewed drafts of the article, and approved the final draft.

- Le Minh Ly performed the experiments, authored or reviewed drafts of the article, and approved the final draft.
- Pham Thi Phuong Thao analyzed the data, prepared figures and/or tables, and approved the final draft.

## Data Availability

The raw measurements are available in the Supplementary File.

## Supplemental Information

Supplemental information for this article can be found online at http://dx.doi.org/10.7717/peerj.17992#supplemental-information.

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
