# Peer review of "Yield-limiting nutrient response of lowbush blueberry grown in recent and ancient alluvial soils of the Mekong Delta"

_PeerJ, doi:10.7717/peerj.17992_

## Round 0.1 · original submission · Major Revisions

Your paper requires major revisions.

**Language Note:** The review process has identified that the English language must be improved. PeerJ can provide language editing services - please contact us at [email protected] for pricing (be sure to provide your manuscript number and title). Alternatively, you should make your own arrangements to improve the language quality and provide details in your response letter. – PeerJ Staff

Reviewer 1 ·

Basic reporting

1.The english language used in the article is not adequate and fails to meet the journal standards.
The incorrect use of grammar at multiple places adds to the ambiguity of the article, making it difficult to understand by the readers

2. The number of references is sufficient and clear background has been provided.

3. Though the article is structured, but the plant pictures shown in the article appear very unscientific and unprofessional.

Experimental design

Research question is well defined and meaningful. The article tries to answer the research question showcasing sufficient scientific evidence but still somehow fails to make an impact.

The random block design experiments are well planned and performed in an adequate manner with satisfactory results, but lack in proper result presentation is a setback.

Methods have been described in proper detail and sufficient information has been conveyed to replicate the results.

Validity of the findings

The findings of the article are indeed valuable and shed light on the productivity difference of recent and ancient alluvial soil of the Mekong Delta region, especially in context to blueberry cultivation.

Experiments to show the nutrients limiting to the blueberry cultivation in both recent as well as ancient alluvial soil are well planned and performed in an adequate manner with satisfactory results, but lack in proper result presentation is a setback.

Conclusion is well stated and supports presented results.

Reviewer 2 ·

Basic reporting

Ngo et al., Yield-Limiting Nutrient Response of Lowbush Blueberry Grown in Recent and Ancient Alluvial Soils of the Mekong Delta

In this study, the authors conducted a potted experiment using two soil types and four nutrient treatments (NPK, PK, NK, and NP) for a lowbush blueberry species (Vaccinium tenellumn). The authors found that the blueberry yield in AS soils was lower compared to that in RA soil, and N omission resulted in reduced foliar N concentrations and reduced yield in both soil types. The results are interesting and are important for lowbush blueberry management. My major concerns are 1) the literature and recommendations on mainly for another blueberry species (Vaccinium angustifolium), whether those published results (e.g. Trevett, 1972) can apply to Vaccinium tenellumn need to be clarified and discussed; 2) The writing needs to be improved in terms of style. Specifically, there are two many short paragraphs with only two sentences. Please use a good published paper for reference and organize the paragraphs better; 3) the grammar also needs to be checked carefully; 4) statistical results need to be added to the figures.

Some detailed comments are listed here:
Line 28-30-Abstract-Results: use % for reduced yield and weight will make more sense.
Line 32 to 33-Abstract-results: define ARE and INS.
Line 38- please provide the scientific name of the blueberry species studied or described here.
Line 48-52: Most cited studies are on Vaccinium angustifolium, and their results may not be able to apply to Vaccinium tenellumn
Line 78-80: This sentence is not complete.
Line 82 to 87: please clarify when the study was conducted.
Line 106: The experiment “was” performed.
Line 106: How large is the pot?
Line 149: which plant parts? Leaves only or other parts like stems?
Line 162: tree or shrub?
Line 170: does N omission mean control (without N fertilizer application)
Figure 3. Vaccinium tellenum is not a genus, it is a species.
Figure 4 and 5: please label the statistics-significant differences.
Line 358: check the grammar of the sentence.
Discussion- The Trevett, 1972 threshold is for another species. Whether it is useful for Vaccinium tenellumn needs to be discussed.

Experimental design

The experiment design was solid, but more details are needed.

Validity of the findings

Statistical results need to be added to the figures.

Additional comments

The writing, style, and grammar need to be improved.

---

## Round 0.2 · Major Revisions

I agree with Reviewer 2 that the authors haven't fully addressed the reviewers' comments yet. While Figures 3 and 5 are new additions, they currently lack the scientific rigor needed to support claims like 'N omission resulted in the lowest plant biomass (Figure 3)'.

However, the experiment itself appears well-designed, and the results have potential significance. Therefore, I recommend offering the authors another chance to revise their paper. Specifically, I suggest they carefully review all comments from both reviewers in both review rounds and provide a detailed response to each point.

Reviewer 1 ·

Basic reporting

The english of the manuscript has been thoroughly revised, adding further impact to the findings of the study.

Experimental design

The planning as well as presentation of the experimental design is unambiguous and clearly understandable.

Validity of the findings

The findings of the experiments are statistically sound and have been clearly presented in easy and scientific language.

The findings are clear, impactful and novel, further supplementing the already published literature in the field.


Conclusion is well written and summarizes the entire study, along with the study background, experimental design and the findings in unambiguous language.

Additional comments

None

Reviewer 2 ·

Basic reporting

Ngo et al., Yield-Limiting Nutrient Response of Lowbush Blueberry Grown in Recent and Ancient Alluvial Soils of the Mekong Delta

I have reviewed the previous version of the manuscript, and am glad to see a revised version. I thank the effort by the authors and think that the manuscript has been improved to some degree. However, the responses are unclear and some of the revisions are not sufficient.

For the response to point 1.1, it mentioned that “The use of Trevett’s range was discussed in line 667-674.” But there are no lines 667-674 in the manuscript. The authors didn’t explain how they revised and discussed the potential issue. Some direct answers (responses) in the response letter will be helpful. In Lines 351-360 of the document with track changes, the authors only edited the writing, and didn’t add anything regarding the potential fitness and/or problems applying Vaccinium angustifolium threshold to Vaccinium tenellum. This needs to be carefully discussed, and the reason why Vaccinium angustifolium threshold could be applied to Vaccinium tenellum needs to be well-explained.

The writing and grammar still need to be improved more. E.g. “no specific threshold range has been found for evaluating leaf nutrient content for Vaccinium tellenum.” It reads like Vaccinium tellenum does not have a threshold range. It does have a threshold range, and it is just that it has not been studied. Here the correct word should be “established”, rather than “found”.

Introduction, the first sentence is exactly the same as the first sentence in Abstract. Please avoid repeating the same sentence. Please rewrite one of the sentences.

Line 42-43: Blueberries have a lot of species and varieties. Some are shade-torelant understory species. So please be specific and precise.

Line 70-72: “and respond differently to crop productivity.” It is that: crop productivity responds to soils with different soil fertility. Not that soil fertility responds to crop productivity.

In the response, the authors state that “Statistical results have been added to the figures of 4 and 6.” However, I still could not see your statistical results in the figures. The figure legends also did not describe the statistical tests used. If there is no significant difference, the authors also need to point it out.

I asked the authors to provide the scientific name of the blueberry species studied at the beginning of the introduction. The authors responded “That is Vaccinium tenellumn”, but never revised it in the manuscript. Blueberries have a lot of species and a lot of varieties. Not all of them are “in pots in many yards in the Mekong Delta (MD) region”. The authors need to improve the preciseness in the writing. Also, the species name is not correct here in the response.

Experimental design

N/A

Validity of the findings

N/A

Additional comments

N/A

---

## Round 0.3 · Minor Revisions

As I mentioned in my last comments, Figures 3 and 5 currently lack the scientific rigor needed to support claims like 'N omission resulted in the lowest plant biomass (Figure 3)'. Please carefully review the discussions related to Figures 3 and 5 to ensure rigorous scientific presentation. All figures and tables must include clear and concise captions explaining statistical results, such as the meaning of asterisks and lowercase letters. These captions should provide sufficient information for readers to fully comprehend the data without referring to the manuscript text.

---

## Round 0.4 · accepted · Accept

Thanks for authors' time revising the manuscript. I belive it now reach the standard for publication. There is a typo in the caption of Figure 5, please revise it during the proofread stage.